# High-throughput screening reveals diverse lambdoid prophage inducers

Gayatri Nair,[1] Rabia Fatima,[1] Alexander P. Hynes[1,2]

**ABSTRACT** Prophages, dormant bacteriophage genomes integrated within the bacterial chromosome, play pivotal roles in shaping microbial communities when awakened. Our current understanding of prophage activation is largely shaped by a narrow set of traditional DNA-damaging inducers, such as mitomycin C and ciprofloxacin, which trigger the bacterial SOS response. This study employed high-throughput screening of 3,921 compounds to identify novel prophage inducers using model lambdoid prophage HK97. We identified multiple new inducers across diverse pharmacological classes, including dietary supplements and therapeutics. Despite the variety in compounds, all acted through SOS-dependent pathways. However, bleomycin, an antineoplastic antibiotic, demonstrated broad-spectrum and potent prophage induction exceeding standard inducers, with activity validated across multiple phage-host pairings. These findings expand the repertoire of prophage inducers into commonly ingested xenobiotics and introduce bleomycin as a powerful, cost-effective tool for prophage research.

**IMPORTANCE** Around 75% of bacteria carry within them dormant viruses (prophages), which can awaken when the bacterium is stressed, killing the bacterium. Historically, this has been done using DNA-damaging antibiotics, but increasingly, more such signals have been discovered. Here, through a high-throughput screen, we identify phage-waking activity in several commonly consumed compounds, such as the SSRI Prozac, as well as a new DNA-damaging agent that is much more effective in waking phages than the previous gold standard.

**KEYWORDS** SOS pathway, HK97, berberine, Prozac, fluoxetine, harmane, mitomycin C, bleomycin, high-throughput, induction

Bacteriophages (phages), the most abundant biological entities on Earth, have their genomes integrated as prophages in 75% of sequenced bacterial genomes (1, 2). Far from being inert, these prophages can profoundly alter bacterial behavior and shape microbial ecosystems when "awakened" (3–5). Despite their prevalence and impact, our understanding of prophage activation—known as induction—has been shaped by a handful of experimental tools. Most studies rely on synthetic agents like mitomycin C and ciprofloxacin, as well as ultraviolet (UV) light, to stimulate prophage induction in model systems like lambda, a temperate phage of *Escherichia coli* (6–8). Exposure to these strongly and reproducibly induces the bacterial SOS response, which in turn triggers prophage induction (8–10). While effective, this approach has become the default across studies, potentially biasing our view of phage–host interactions and limiting the discovery of alternative inducers that may be more relevant to natural environments or clinical contexts.

Compared to other environments, the human microbiome is especially enriched for temperate phages, which predominantly exist in a lysogenic state by integrating

Address correspondence to Alexander P. Hynes, hynesa3@mcmaster.ca.

The authors declare no conflict of interest.

See the funding table on p. 10.

*[This article was published on 27 October 2025 with an error in Fig. 4. The figure was corrected in the current version, posted on 5 November 2025.]*

their genomes into the host chromosome and persisting as prophages (11–13). These prophages replicate passively with the host cell until specific stimuli induce the switch to the lytic replication cycle, resulting in bacterial lysis and the release of new virions (14). In lambdoid prophages, such as λ and HK97, this switch is governed primarily by the CI repressor. This induction is typically associated with the bacterial SOS response —an inducible DNA repair system regulated by the LexA repressor and the RecA inducer, which stimulates autocatalytic cleavage of CI allowing for transcription of early lytic genes (15–17). While DNA damage is the best-characterized trigger for prophage induction, phages would benefit from a more complete picture of the health of the host cell, especially when competing with other prophages within a polylysogen. Such triggers have been overlooked by SOS-focused studies.

Previous studies have identified various foods and compounds capable of inducing prophages. For example, Oh et al. reported that *Lactobacillus reuteri* prophages can be induced with dietary fructose and short-chain fatty acids via RecA-dependent activation of the Ack pathway (18). Similarly, Silpe et al. demonstrated that colibactin, a genotoxin produced by certain gut microbiome members, triggers prophage induction in *Salmonella* via the SOS response (19). While these studies explored mechanistic links between specific compounds and phage activation, they focused on a narrow set of inducers.

However, broader screens have begun to emerge. For example, Boling et al. assessed 117 foods and found that compounds such as artificial sweeteners and bee propolis increased virus-like particle (VLP) counts and inhibited bacterial growth in a species-specific manner (20). Validation of selected hits was limited to VLP quantification by flow cytometry—a method that cannot distinguish between infectious and non-infectious particles. Similarly, Tompkins et al. screened 3,747 compounds using a high-throughput luminescent β-galactosidase assay which led to the identification of four naturally derived compounds as potential prophage inducers (21). However, the method used measures phage gene expression and not actual phage production or infectivity. Other studies have more broadly examined the effects of commonly consumed compounds on the gut microbiome. Sutcliffe et al. tested 12 medications at five different concentrations against eight bacterial isolates; yet the methods of validation faced similar limitations to those described above (22).

These studies highlight that diverse environmental and dietary agents can modulate bacterial and phage dynamics. However, the extent to which frequently encountered compounds broadly influence prophage induction remains largely unexplored. Systematic, functionally validated approaches—particularly those confirming infectious phage production—are essential for uncovering overlooked interactions and expanding our current understanding of phage–host dynamics. Addressing this gap will provide researchers with new tools to broaden the study of prophage biology, enabling investigation into alternative induction pathways, particularly in cases where compounds are not traditionally associated with DNA damage.

Here, we employ high-throughput screening (HTS) using a large compound library to identify compounds capable of triggering prophage induction in model lambdoid prophage HK97 (23). When lysogenic bacteria are exposed to compounds capable of inducing the lytic cycle, phage-mediated lysis may occur, resulting in reduced endpoint optical density (OD) at concentrations that would not necessarily affect non-lysogenic counterparts (24). This differential growth phenotype provides a scalable and accessible readout for HTS. By comparing the growth dynamics of lysogenic and non-lysogenic strains, candidate inducers can be identified and subsequently validated through direct phage quantification. This approach enables systematic discovery of previously unrecognized prophage inducers and lays the groundwork for exploring alternative induction pathways beyond canonical DNA damage responses.

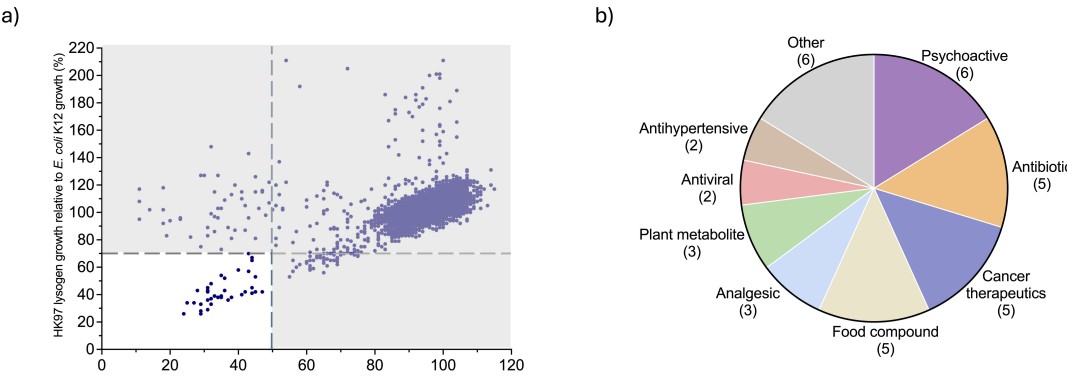

**FIG 1** High-throughput screening identified 37 primary hits as potential inducers. (a) Primary hits identified in the HK97 lysogen model. Each dot represents a compound, and hits are defined as compounds that, averaged across both replicates, cause 50% or less growth relative to a no compound control and 70% or less growth relative to *E. coli* K12 HER1382 + compound. These are represented in the white box. (b) Categorization of primary hits.

## RESULTS AND DISCUSSION

### HTS identified several novel prophage inducers

We conducted a primary screen using *E. coli* K12 HER1382 and its HK97 lysogen exposed to 3,921 bioactive compounds to identify compounds capable of inducing prophage-mediated bacterial killing. Ciprofloxacin at MIC (minimum inhibitory concentration) served as a positive control for growth inhibition and $z'$ factors were calculated to ensure assay quality (Table S2). While media-only negative controls showed no growth and cell-only conditions exhibited no inhibition, the ciprofloxacin MIC control resulted in growth inhibition of both strains, reducing growth to 18% (HK97 lysogen) and 20% (*E. coli* K12 HER1382) relative to untreated controls, as expected. Because the ciprofloxacin control was added at MIC of the more resistant (non-lysogen) strain, it should not differentially affect the two strains despite the known inducing properties of the antibiotic.

Compounds were designated as primary hits if exposure of HK97 lysogen to the compound resulted in less than 50% bacterial growth relative to the untreated control on the same plate, and if growth of the HK97 lysogen was 70% or less compared to *E. coli* K12 HER1382 treated with the same compound. These criteria were selected to enrich for compounds that cause increased bacterial killing in the presence of the prophage. Application of these thresholds yielded 37 primary hits (Fig. 1a, Table S3). These compounds produced a greater killing effect in the presence of both the compound and the prophage, consistent with the sensitization we expect from prophage induction. A single compound, benzoxyquine, resulted in a massive benefit to the lysogen—sitting at coordinates 97, 291, and was omitted from Fig. 1a to increase legibility.

Notably, several known SOS pathway inducers such as fluoroquinolones lomefloxacin hydrochloride and ofloxacin were identified among the hits, lending support to the validity of the screen (25, 26). However, the identified compounds spanned a wide variety of pharmacological classes, including antivirals, antineoplastics, analgesics, and dietary supplements (Fig. 1b), suggesting that selection was not restricted to traditional antibiotics. This diversity highlights the potential to identify novel prophage inducers. However, because hits were identified based on growth inhibition, which was used as a proxy for prophage induction, we carried out further validation to confirm whether these compounds directly trigger prophage induction.

To validate selected primary "hits," 13 compounds were ordered for further analysis, based on a combination of cost and functional diversity: valproic acid, thiothixene HCl, chlorambucil, chlorophyllide Cu complex Na salt, astaxanthin, gibberellic acid, docosanol, oxoantel pamoate, imiquimod, berberine, fluoxetine HCl, harmane, and bleomycin (Fig. 2a). Of these 13, 8 failed to yield any MIC in our larger volume assays (Fig. S1). For

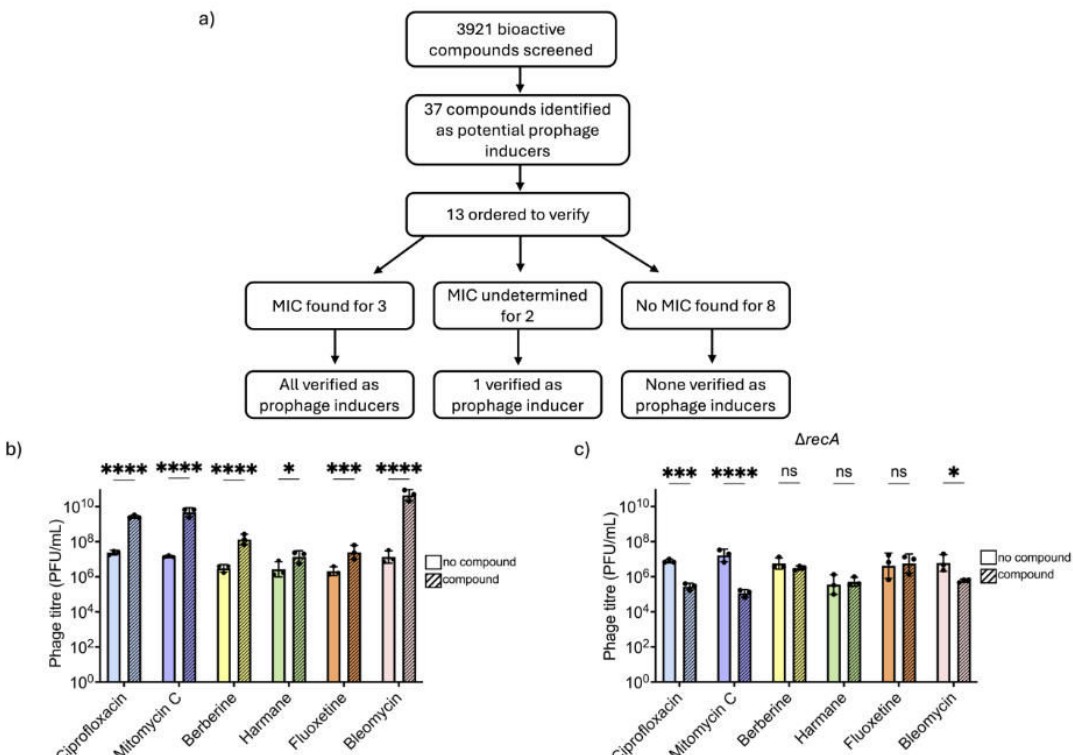

**FIG 2** Characterized primary hits are SOS-dependent prophage inducers. (a) Summary of the validation process of ordered hits. MIC is defined as the minimal inhibitory concentration. (b) Phage quantification of HK97 lysogen filtrates alone and challenged with different compounds across three biological replicates. (c) Phage quantification of ΔrecA lysogen lysates alone and challenged with different compounds across three biological replicates with each black dot representing a replicate. Error bars represent SD. Titers compared using two-way ANOVA and Šidák's multiple comparison tests. *$P \leq 0.05$, **$P \leq 0.01$, ***$P \leq 0.001$, and ****$P \leq 0.0001$. ns = non-significant.

two more, the MICs could not be determined due to compound precipitation and pigmentation. However, berberine, harmane, fluoxetine HCl, and bleomycin were verified as prophage inducers (Fig. 2b).

Berberine, a plant alkaloid with weak antibiotic activity, is the primary bioactive component of tree turmeric and has been traditionally used in Ayurvedic and Chinese medicine (27). It has also been shown to modulate metabolic disorders, including type 2 diabetes and dyslipidemia, through AMPK activation and enhancement of glycolysis, leading to reduced insulin resistance (28, 29). MIC could not be determined, as compound pigmentation and precipitation interfered with spectrophotometric assays (Fig. S1). Regardless, the lysogen was subjected to a range of berberine concentrations, and the greatest magnitude of induction (~2-log increase) relative to baseline spontaneous induction levels was observed at 350 µg/mL via plaque assay (Fig. 2; Fig. S2).

Harmane, a heterocyclic β-carboline, is found in commonly consumed products such as cooked meats, coffee, soy sauce, and tobacco products (30). It functions as a monoamine oxidase inhibitor and has been linked to altered dopamine metabolism (31, 32). Harmane MIC for *E. coli* K12 HER1382 was found to be 100 µg/mL and exposure of HK97 lysogen to MIC resulted in approximately 1-log increase in phage titer relative to baseline spontaneous induction levels (Fig. 2; Fig. S1 and S2, Table S1).

Fluoxetine HCl (Prozac), a selective serotonin reuptake inhibitor (SSRI), is widely prescribed for the treatment of mood disorders and is among the most frequently used antidepressants in Canada, where approximately 17% of the population is prescribed antidepressants (33, 34). Fluoxetine MIC for *E. coli* K12 HER1382 was found to be 100 µg/mL and exposure of HK97 lysogen to ½ MIC resulted in approximately 1-log increase in phage titer relative to baseline spontaneous induction levels (Fig. 2; Fig. S1

and S2; Table S1). Notably, berberine, fluoxetine, and harmane have all been reported to impact the gut microbiome and/or exhibit antibacterial properties, suggesting a potential role for prophages in mediating their effects (35–37). Although none of these compounds are currently recognized as prophage inducers, recent evidence indicates that fluoxetine can activate the bacterial SOS response in *E. coli*, resulting in Shiga Toxin production which stx phages are known to mediate (38, 39). These findings further emphasize the likelihood that induced phages may play an existing role in altering microbiome composition, influencing the therapeutic effects of these compounds. Moreover, the identification of these xenobiotics as prophage inducers highlights their potential utility as tools for probing phage-host dynamics and understanding how non-antibiotic compounds influence microbial communities in both clinical and environmental contexts.

Finally, bleomycin is a glycopeptide antibiotic primarily used as an antineoplastic agent to treat squamous cell cancer, Hodgkin's lymphoma, and testicular carcinoma (40). Its activity as a prophage inducer in *E. coli* and *Bacillus subtilis* had been reported in the 70s, considered potent, but could not be readily compared to gold standards due to its low purity (41)—although its role in induction was further characterized (42). In the 80s, across a narrow range of concentrations tested, it appeared to reach slightly higher induction values than mitomycin C (43). In our system, bleomycin MIC for *E. coli* K12 HER1382 was found to be 6.25 µg/mL and exposure of HK97 lysogen to ½ MIC resulted in a greater than 3-log increase in phage titer relative to baseline spontaneous induction levels (Fig. 2b; Fig. S1; Table S1). This exceeds induction levels observed with ciprofloxacin and mitomycin C. While the compound's ability to induce prophage is established, the effectiveness and exceptionally high induction had not been remarked upon or delved into (44).

Induction by all four compounds was dependent on the SOS response, as no induction was observed in an HK97 Δ*recA* lysogen where the SOS pathway is disabled (Fig. 2c). Note that as the same concentrations are displayed as for Fig. 2a, and the *recA* strains are often more susceptible to DNA-damaging compounds, we attribute the decreased titres for ciprofloxacin and berberine to (phage-independent) impeded growth of the lysogen. These findings validate that the compounds for which MICs could be obtained are prophage inducers, thereby supporting the overall robustness of the screening strategy.

Physiological concentrations of these compounds vary depending on dosage, metabolic rate, time since administration, and route of delivery. For berberine, a widely used dietary supplement, the recommended daily dose is approximately 1,500 mg throughout the day, where peak plasma levels reach 0.4 ng/mL after a single dose of 400 mg, substantially lower than the concentrations used in our assays (45). Similarly, harmane levels are difficult to estimate accurately, given its presence in a range of commonly consumed foods such as meats, coffee, and tobacco products. In a population-based study, the median blood concentration in controls was very low ($2.7 \times 10^{-10}$ g/mL) after habitual dietary intake (46). With both berberine and harmane, local concentrations in the gastrointestinal tract are likely to be substantially higher than systemic levels due to the prevalence of harmane in common diets and, for berberine, the frequent supplement use, lack of standardized dosing, and potential for repeated high dose exposure. Fluoxetine, at a maximum recommended daily slow-release dose of 90 mg, reaches plasma concentrations around 91–302 ng/mL (0.091–0.302 µg/mL), also lower than our assay concentrations (47). In contrast, bleomycin achieves plasma levels between ~0.9 and 2.2 µg/mL which overlap with the concentrations used in our screen (48). Unlike the other compounds identified, bleomycin is a cytotoxic antibiotic and is unlikely to be used intentionally to modulate the microbiome. However, consistent with earlier reports, the magnitude of prophage induction observed with bleomycin exceeds that of ciprofloxacin and mitomycin C—both well-established, SOS-dependent inducers—highlighting its potential as a powerful tool for studying prophage induction.

## Characterization of bleomycin as an alternative to standard prophage inducers

To further characterize bleomycin activity, HK97 lysogen was exposed to a range of bleomycin concentrations (400–0.0003 µg/mL). For comparison, the same was done with ciprofloxacin and mitomycin C. Ciprofloxacin induced at least a 1-log increase in phage titers relative to baseline spontaneous induction at concentrations between 0.25L and 0.004 µg/mL (Fig. 3). Mitomycin C displayed broader and more potent activity with at least 1-log induction across concentrations 25–0.002 µg/mL (Fig. 3). Compared to ciprofloxacin and mitomycin C, bleomycin induces at an overall higher magnitude across a broader concentration range with at least a 1-log induction at concentrations 200–0.002 µg/mL (Fig. 3). Despite the strong prophage induction observed with bleomycin, this effect does not correlate with magnitude of SOS activation, measured by *recA* expression, which was higher in ciprofloxacin (Fig. S3). Interestingly, the broad range of bleomycin concentrations that induced phage production corresponded with a relatively stable level of *recA* expression across the same tested concentrations apart from a curious spike at 0.25 µg/mL (Fig. S3).

Traditional inducers like ciprofloxacin and mitomycin C often require dose optimization and laborious troubleshooting due to their limited range and magnitude of activity. In contrast, bleomycin's broader effective concentration range and higher magnitude of induction reduce the need for optimization, serving as a more flexible alternative. To determine whether this represents a generalizable phenomenon, we tested bleomycin's ability to induce other phages, including in other bacteria.

To investigate this, we first exposed phage Mu lysogens to a range of ciprofloxacin, mitomycin C, and bleomycin concentrations. Mu is a temperate transposable phage that infects *E. coli* 40 by transposing its DNA into the host chromosome, almost at random (49). While no chemical treatments have previously been reported to induce Mu, induction has been achieved through specific viral mutants (50). Here, three Mu lysogens were used, each characterized by distinct integration sites confirmed via arbitrary PCR. Unexpectedly, exposure of Mu lysogens to ciprofloxacin, mitomycin C, and bleomycin was able to induce Mu in select lysogens. All three compounds were able to induce lysogen 3, but the effect was lost completely in lysogen 2 and partially in lysogen 1 (Fig. 4a). This suggests that Mu is inducible under certain genomic contexts with host integration potentially influencing sensitivity.

To determine if bleomycin's activity extended to other bacterial species which have yet to be explored, we tested its effects along with ciprofloxacin and mitomycin C on three *Pseudomonas aeruginosa* PA14 lysogens, each harboring Mu-like phages related to known *Pseudomonas* transposable phages (51). In contrast to the broader induction profiles seen with HK97, phage induction in *P. aeruginosa* was limited, with induction typically occurring at a single concentration for each compound. This occurred near the MIC for ciprofloxacin of 0.063 µg/mL and near the MIC for mitomycin C (0.125–0.5 µg/mL). Bleomycin MIC was undeterminable with little reduction in OD at the highest concentration tested, yet induction was readily observed at 200 µg/mL. Ciprofloxacin and mitomycin C induced only phage Drowsy and Hali, while bleomycin induced all

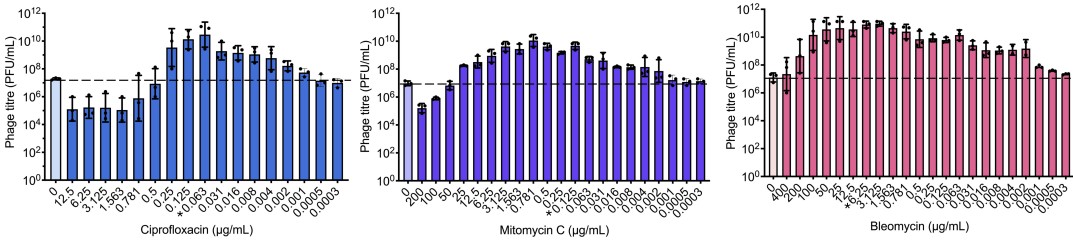

**FIG 3** Bleomycin demonstrates greater activity compared to ciprofloxacin and mitomycin C. Phage quantification of HK97 lysogen either alone or challenged with ciprofloxacin, mitomycin C, or bleomycin across a range of concentrations, averaged across three biological replicates with each black dot representing a replicate. * Denotes MIC. Dotted line represents baseline induction levels. The bars represent the means, and the error bars represent the standard deviations.

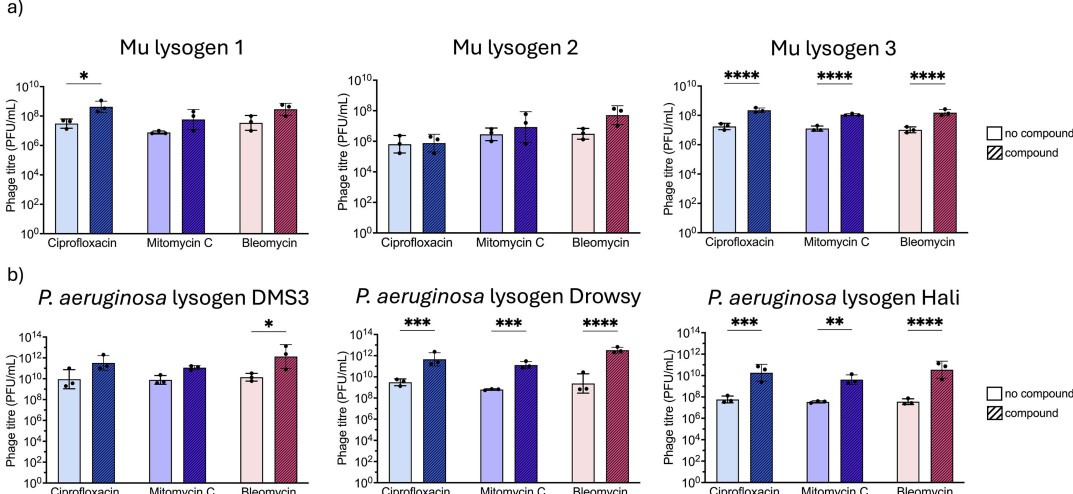

**FIG 4** Effects of bleomycin are generalizable across phages and bacterial hosts. (a) Phage quantification of *E. coli* phage Mu lysogen lysates alone and challenged with ciprofloxacin, mitomycin C, or bleomycin, averaged across three biological replicates. Lysogen number corresponds to the phage integrating at a different site within the host genome. (b) Phage quantification of different *P. aeruginosa* lysogen lysates alone and challenged with ciprofloxacin (0.063 µg/mL), mitomycin C (0.125–0.5 µg/mL), or bleomycin (200 µg/mL), averaged across three biological replicates with each black dot representing a replicate. Error bars represent SD. Titers compared using two-way ANOVA and Šidák's multiple comparison tests. *$P \leq 0.05$, **$P \leq 0.01$, ***$P \leq 0.001$, and ****$P \leq 0.0001$.

three lysogens and resulted in higher magnitudes of induction, further emphasizing the compound's potency and suggesting generalizability across different bacterial host and their prophages, as has been demonstrated for ciprofloxacin and mitomycin C (Fig. 4B).

## Conclusions

This study employed HTS to identify several novel, commonly consumed compounds as prophage inducers, reinforcing the concept that phage induction is a key mechanism by which external agents can shape microbial community dynamics. Although most compounds were tested at supra-physiological concentrations, this does not preclude their capacity to act as prophage inducers, especially in the gut, where localized exposure may be higher or more prolonged. The identification of widely used therapeutics and dietary supplements as inducers supports growing evidence that non-antibiotic agents can inadvertently modulate microbial ecosystems through phage-mediated effects (18–22). Notably, all validated inducers operated via the canonical SOS response, underscoring the conserved nature of this pathway.

Among these compounds, bleomycin emerged as a potent and broadly effective prophage inducer, often outperforming current standard inducers ciprofloxacin and mitomycin C (6, 7). In addition to this performance, bleomycin is also more cost-effective and chemically stable compared to mitomycin C, making it well-suited for routine laboratory use (52). Together, these findings highlight the power of high-throughput approaches in identifying prophage inducers and establish bleomycin as a valuable tool for prophage research with the potential to replace currently used prophage inducers.

## MATERIALS AND METHODS

### Bacterial strains

All bacterial strains are grown at 37°C shaking at 130 rpm or 250 rpm in LB broth which could refer to Table 1.

**TABLE 1** Bacterial and viral strains used in this study

| Strain | Source |
|---|---|
| *E. coli* K12 HER1382 | Félix d'Hérelle Reference Center for Bacterial Viruses |
| *E. coli* K12 HER 1382 HK97 lysogen | Al-Anany et al. (53) |
| *E. coli* BW25113 Δ*recA* | Dharmacon KEIO collection through Horizon Discovery (Cambridge, UK) |
| *E. coli* BW25113 Δ*recA* lysogen | This work |
| *E. coli* 40 | Félix d'Hérelle Reference Center for Bacterial Viruses |
| *E. coli* 40 Mu lysogen 1; integrates near *ulaD* gene | This work |
| *E. coli* 40 Mu lysogen 2; integrates near *yhaO* gene | |
| *E. coli* 40 Mu lysogen 3; integrates in *ygbM* gene | |
| *P. aeruginosa* PA14 | Burrows Lab, McMaster University |
| *P. aeruginosa* PA14 lysogen Drowsy | Fatima et al. (51) |
| *P. aeruginosa* PA14 lysogen DMS3 | Fatima et al. (51) |
| *P. aeruginosa* PA14 lysogen Hali | Fatima et al. (51) |
| *E. coli* K12 MG1655 | Shao et al. (54) |
| *E. coli* K12 MG1655 *recA*–GFP | Zaslaver et al. (55) |
| *E. coli* K12 MG1655 *sulA*–GFP | Zaslaver et al. (55) |
| *E. coli* K12 MG1655 *pitB*–GFP | Zaslaver et al. (55) |

## Generating HK97 Δ*recA* lysogen

To generate the HK97 Δ*recA* lysogen, serial dilutions of HK97 phage were plated on a soft agar overlay (10 mL 1% LB + 3 mL 0.75% LB with 300 µL of culture) of Δ*recA* and incubated overnight at 37°C. Regrowth was re-streaked, incubated overnight, and 32–48 colonies were picked and purified. Each purified colony was then streaked on top of a soft overlay containing parental (phage-sensitive) Δ*recA* strain, using a HK97 lysogen as a positive control. Plates were incubated overnight, and HK97 Δ*recA* lysogens were identified by the presence of zones of clearing surrounding the streak.

HK97 lysogen confirmation was done using PCR primers attBF and HK97lysR detecting the known phage-host junctions. The absence and presence of the *recA* gene was verified by PCR using primers binding both within (RecA_Fwd and RecA_Rev) and outside (RecA_F and RecA_R) the gene and further validated by whole-genome sequencing. The primers are available in (Table 2).

## Prophage induction assays

Induction assays were performed in 96-well plates using two-fold serial dilutions of the compound and incubated at 37 °C for 18 h. $OD_{600}$ readings were taken at endpoint (18 h). Cultures were added at an initial OD of 0.2, prepared by a 1:100 sub-inoculation from an overnight culture. Each well had a final volume of 250 µL. To confirm and test the inducibility of lysogens, ciprofloxacin and mitomycin C were used as positive controls. MIC and ½ MIC of compounds were determined using the parent bacterial host and applied to the corresponding lysogens. When the MIC could not be determined, induction assays were done using the highest compound concentration permissible,

**TABLE 2** Primers used in this study

| Primer name | Sequence |
|---|---|
| attBF | TGAATCCGTTGAAGCCTGCT |
| HK97lysR | GCGTGTAATTGCGGAGACTT |
| RecA_Fwd | CGGTATTACCCGGCATGACA |
| RecA_Rev | GCAGATGCGACCCTTGTGTA |
| RecA_F | GTCAACCAGTTCGCCGTAGA |
| RecA_R | GGGCCGTATCGTCGAAATCT |

**TABLE 3** Compounds used in this study

| Compound | Solvent | Source |
|----------|---------|--------|
| Berberine (chloride) | DMSO | Cayman chemical company; Cat #10006427 |
| Bleomycin sulfate | DMSO | Cayman chemical company; Cat #13877-50 |
| Ciprofloxacin | PCR $H_2O$ | Cayman chemical company; Cat #14286-5 |
| Fluoxetine HCl | DMSO | Cedarlane; Cat #14418-5 |
| Harmane, 98% | DMSO | Sigma-Aldrich; Cat #103276 |
| Mitomycin C | PCR $H_2O$ | Cedarlane Labs; Cat #SIH-246 |

considering solubility and the percentage of organic solvent used. Compounds were either dissolved in PCR $H_2O$ or dimethyl sulfoxide (DMSO). In instances where DMSO was used, vehicle controls confirmed the solvent did not have any effect on phage titer (Fig. S4). The source of all compounds is detailed in (Table 3).

After incubation, plates were either filtered (MilliporeSigma Multiscreen High Volume Filter Plates: 0.45 µM) or centrifuged (2103 RCF, 25 min). Approximately 100 µL of the resulting supernatant was aspirated, serially diluted (10-fold) in LB broth, and plated (3 µL) on respective bacterial hosts (*E. coli* K12 HER1382 Δ*recA*, *E. coli* K12 HER1382, and *E. coli* 40, PA14). When working with PA14 bacterial strains, biofilms were removed prior to centrifuging by inserting filter tips into the wells and allowing the biofilm to adhere. Phage concentration is calculated via the following formula:

$$PFU/mL = (\text{\# of plaques / volume spotted}) * \text{dilution factor}$$

## HTS

The HTS was an integrated system with most operations occurring offline, requiring little manual assistance. Each assay plate was barcoded prior to the experiment.

A total of 3,921 bioactive compounds from the Center for Microbial Chemical Biology (CMCB, McMaster University, Hamilton, Ontario, Canada) collection were tested against two bacterial strains (*E. coli* K12 HER1382 and *E. coli* HK97 lysogen) using 384-well plates. Each assay plate contained one bacterial strain grown to an OD of 0.4. The HTS pipeline involved dispensing the bioactive compounds in duplicates at a final concentration of 10 µM with 30 µL of culture and 0.3 µL of compound added using the Echo Acoustic Liquid Handler (Beckman Coulter, CA, USA), followed by the addition of controls—DMSO (negative control) and ciprofloxacin (positive control)—using the Multidrop Combi Reagent Dispenser (Thermo Fisher Scientific, MA, USA). Subsequently, 30 µL of culture was added using the Tempest Liquid Dispenser (Formulatrix, Dubai, UAE). Plates were incubated at 37 °C for 6 h, with OD readings taken every 30 min using the Biotek Epoch Plate Reader (Agilent, CA, USA). Each assay plate included a column of DMSO (1%) (compounds are diluted with DMSO) + LB media (cells are grown in LB), a column of DMSO (1%) + MIC ciprofloxacin + LB media, a column of DMSO (1%) + cells and a column of DMSO (1%) + MIC ciprofloxacin + cells. These conditions served as negative and positive controls. Screening was performed for a total of 6 h, with reads taken every 30 min.

### Data analysis

Percent growth relative to the average of each plate's untreated controls was calculated individually for each plate and then averaged across the two technical replicates. $OD_{600}$ readings obtained at 6 h were used for analysis.

## SOS gene expression assay

To quantify the magnitude of SOS pathway activation, three *E. coli* MG1655 strains containing engineered promoter-reporter gene constructs were used, each expressing GFP in response to activation of *recA*, *sulA*, or *pitB* expression. SulA was chosen to

represent genes far deeper in the SOS pathway (56), while *pitB* was selected as it was known to be SOS-independent (57), and has been used as a control for similar reporter gene assays by other groups (58). Assays were done in 96-well plates with two-fold dilutions of compound added to 100 µL of culture grown to OD 0.2. Plates were incubated at 37°C for 18 h, after which absorbance and fluorescence reads (479 and 520) were obtained. Strains were maintained in the presence of kanamycin (50 µg/mL) in overnight cultures to ensure plasmid selection. Resulting fluorescence values are normalized to OD, then further normalized to the fold change observed in *pitB* expression. *E. coli* MG1655 (WT) was used as the negative untreated control. Assays were done in biological duplicates, each with technical duplicates.

## Statistical analysis

All figures and statistical analyses were generated using GraphPad Prism version 10.3.1 (GraphPad Software, Inc., CA, USA). Flowcharts were created using Microsoft PowerPoint.

## AUTHOR AFFILIATIONS

[1]Department of Medicine, McMaster University, Hamilton, Canada

[2]Department of Biochemistry and Biomedical Sciences, McMaster University, Hamilton, Canada

## AUTHOR ORCIDs

Alexander P. Hynes  http://orcid.org/0000-0002-7058-6006

## FUNDING

| Funder | Grant(s) | Author(s) |
|---|---|---|
| Natural Sciences and Engineering Research Council of Canada | 2018-05996 | Alexander P. Hynes |

## AUTHOR CONTRIBUTIONS

Gayatri Nair, Conceptualization, Data curation, Investigation, Methodology, Project administration, Validation, Visualization | Rabia Fatima, Investigation, Validation | Alexander P. Hynes, Conceptualization, Supervision

## ADDITIONAL FILES

The following material is available online.

### Supplemental Material

**Data Set S1 (Spectrum01707-25-s0001.xlsx).** All raw data used to generate figures and tables.

**Fig. S1 (Spectrum01707-25-s0002.tif).** MIC curves of ordered primary hits.

**Fig. S2 (Spectrum01707-25-s0003.tif).** Berberine, harmane, and fluoxetine HCl have narrow range of prophage induction.

**Fig. S3 (Spectrum01707-25-s0004.tif).** Exposure to bleomycin results in consistent levels of SOS induction across a concentration range normalized normalized to pitB-GFP.

**Fig. S4 (Spectrum01707-25-s0005.tif).** Solvent used to dissolve bioactive compounds has no effect on induction.

**Supplemental material (Spectrum01707-25-s0006.docx).** Legends for Fig. S1 to S4; Tables S1 to S3.

## Open Peer Review

**PEER REVIEW HISTORY (review-history.pdf).** An accounting of the reviewer comments and feedback.

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
