## [Reviewer comments · Microbiology Spectrum]

Microbiology Spectrum

High-throughput screening reveals diverse lambdoid prophage inducers

Gayatri Nair, Rabia Fatima, and Alexander Hynes

Corresponding Author(s): Alexander Hynes, McMaster University

Review Timeline:

Submission Date:	June 2, 2025
Editorial Decision:	August 1, 2025
Revision Received:	August 18, 2025
Editorial Decision:	September 13, 2025
Revision Received:	September 16, 2025
Accepted:	October 1, 2025

Editor: Diana Pires

Reviewer(s): The reviewers have opted to remain anonymous.

Transaction Report:

DOI: <https://doi.org/10.1128/spectrum.01707-25>

Re: Spectrum01707-25 (High-throughput screening reveals diverse lambdoid prophage inducers)

Dear Dr. Alexander P Hynes:

Thank you for the privilege of reviewing your work. Below you will find my comments, instructions from the Spectrum editorial office, and the reviewer comments.

The reviewers considered that your manuscript is very interesting but requires additional work. Please return the manuscript within 60 days; if you cannot complete the modification within this time period, please contact me. If you do not wish to modify the manuscript and prefer to submit it to another journal, notify me immediately so that the manuscript may be formally withdrawn from consideration by Spectrum.

Revision Guidelines

Sincerely,
Diana Pires
Editor
Microbiology Spectrum

Reviewer #1 (Comments for the Author):

The manuscript "High-throughput screening reveals diverse lambdoid prophage inducers" shows the discovery and characterization of new potential prophage inducers (lambda-like). Overall, the manuscript is well organized with a clearly defined goal, which was the use of HTS to discover new prophage inducer compounds. Nevertheless, there are some questions/suggestions that were raised while reviewing the manuscript:

- Fig1-a) it is curious that a great portion of the compounds caused an increased relative growth on the HK97 lysogen. Do the authors have an explanation for this? It would be interesting to further explore these differences, which are apparently caused by the presence of this prophage.
 - In the $\Delta recA$ mutant, it would be expected to have the same phage titer as the vehicle control. However, there is a different tendency with the compound treated culture yielding less phages than the no compound-treated. What would be the cause for this?
 - In lines 235 and 239 there is no reference to the physiological concentrations of Harmene and Bleomycin respectively. Despite the difficult estimation that the authors mention, it would be helpful to have some value to make it easier to compare with the results of this study. The authors also say that Bleomycin achieved "plasma levels". If these "levels" overlap with the concentration used, then they should definitely be mentioned.
 - Why did the authors use pitB for mRNA expression assays? And why use sulA as well?
 - In Fig4 there is no identification of the columns in the graphs (there are 2 per treatment)
 - In line 333 is referred P8 and P13 lysogens, but there is no reference to them in the graph
- Small details:
- Table legends end without dot "."
 - In line 386 "OD" misses the wavelength used
 - Figure s2-b) "recA" in the graph should be italic

Reviewer #2 (Comments for the Author):

This study reports the results of a high throughput screen of 3,921 compounds in an attempt to identify novel inducers of a typical *E. coli* lambdaoid prophage, HK97. This prophage has a canonical SOS-inducible cl -like repressor. The screen used differential growth inhibition of the lysogen by the tested compounds compared to an isogenic prophageless control strain, as a proxy readout of potential prophage inducers.

37 potential inducers were identified by the screen, and 13 were selected for further studies. Of these only 4 were verified as prophage inducers. Perhaps not surprisingly, all were dependent on the SOS response, showing no induction in a *recA* mutant background.

Further characterization of the most potent prophage inducer, bleomycin, revealed a broader range and more potent induction than mitomycin C. Additionally, bleomycin was shown to be capable of prophage induction in certain uncharacterized μ lysogens of *E. coli* and three different *Pseudomonas aeruginosa* lysogens.

The results are clear and generally well presented, but attention to the following points would result in a much improved manuscript.

- 1) The introduction needs a clear description of how induction occurs in lambdaoid prophages via the cl repressor - this is completely lacking in the current manuscript. Also no reasoning is presented as to why compounds other than SOS inducers might be expected to induce the prophage.
- 2) While figure 1b presents a pie chart of the type of compounds identified as primary hits - the majority of which proved not able to induce prophage - there are no details of the chemistry or variety of 3,921 compounds included in the initial screen.
- 3) Much previous research on identification and use of bleomycin as prophage inducer is downplayed, incorrectly suggesting only a single previous study (ref. 41, Elespuru and White, 1983). A Pubmed search for "bleomycin prophage" results in 14 hits, although all before 1986, studies from multiple different groups and covering not only *E. coli* and lambda, but prophages in *Bacillus subtilis* and *Staphylococcus*. A 1972 study abstract (article in Italian) indicated bleomycin to be more effective than mitomycin C at inducing lambda prophage.
- 4) Consideration of serum concentrations of compounds as physiological effective concentrations is not very relevant to this in vitro laboratory assay development. Consider removing this.
- 5) This study proclaims the novel identification of xenobiotics as prophage inducers, yet does not do any further characterization of these compounds or expand on their potential utility, focusing instead on more detailed characterization of a known lambdaoid prophage inducer.

More minor points that should be addressed:

Description of methods need clarification

Strains - it is not clear if the two *E. coli* strains are isogenic - i.e. if the HK97 lysogen is directly derived from HER1382; if not this needs stating.

Induction Assays. At what wavelength were OD readings taken? This is subsequently implied to be 600nm in the brief data analysis paragraph. Not clear why or where assay readings were taken every 30 mins or only at endpoint (endpoint only for SOS assays?)

HTS screening. Please provide more details on the selection of the bioactive compounds - is this an off the shelf set produced by CMCB for other purposes? Clarify assay description - it appears that compounds were first dispensed as 1 mM solution then 100x volume of culture was added - I had to reread this section multiple times to figure this out.

Results and discussion

Figure 1. Not mentioned here are two also interesting groupings of compounds - 10 in the left-most region of upper left quadrant that appear to indicate compounds toxic to the lysogen. And a group of 30 or so in the upper right quadrant, that like benzoxyquinone, appear to be beneficial to the lysogen. More could be made of the potential of this assay to identify interesting compounds in prophage biology beyond inducers.

Figure 2. It is not clear or stated in the legend at what dose the compounds in b) and c) are used. Additionally, I am confused by the presence of multiple different no compound controls producing what appear to be significantly different phage titers.

Comment on this variability and its potential effect.

Figure 3. These are very busy graphs. Small dots are individual assays of the triplicate assays - not detailed? X axes are very crowded - consider removing unnecessary decimals for higher doses (we do not need 200.000 for 200 or 50.00 for 50 e.g.)

Figure 4. Again, discuss the variation in no compound controls across experiments as an uncontrolled variable. Compound concentration needs explicitly stating. Temper discussion of "novelty" of bleomycin induction in non lambda and non-E. coli organisms with reference to above cited PubMed articles for other phage and bacteria.

Figure S1. Explain the use of a SulA reporter strain. Edit legend to show control is not MG1655 but a pitB-GFP fusion?

Figure S3 - explain anomaly of MIC levels shown versus figure 3 - here Cipro is 2.5 µg/ml (0.063 in fig 3) Bleomycin is 200 µg/ml (3.125 in fig 3).

Table S2. Also identify other compounds (another color?) selected that did not induce prophage on subsequent testing

Reviewer #3 (Comments for the Author):

Nair et al present a very interesting analysis of bleomycin as an SOS-dependent prophage inducer. There are some low effort additions and/or new analyses that should be made, whether for clarity, completion, or just increasing the strength of the paper. It would especially benefit from trivial sequencing and analysis to determine Mu-like integration sites in the lysogens shown and their potential roles in induction or lack thereof. Together, this is an impressive manuscript that just requires a little more work.

A more thorough review of the literature is needed. There is no mention of previous work on bleomycin effects on phages, which is a known inducer though not commonly used. A quick Google search finds several that were not mentioned. Are your results consistent with previous findings? The conclusions that bleomycin should be used as a standard phage inducer in the lab would be better supported with greater literature analysis. Is your work and conclusion unique from these prior analyses?

Fig 2A: You say that "Regardless, lysogen was subjected to a range of Berberine concentrations and the greatest magnitude of induction (~2-log increase) relative to baseline spontaneous induction levels was observed at 350 µg/mL (Figure 2b)." You should plot the berberine concentration vs prophage induction (as you do for bleomycin in Fig 3). It would be relevant to add the same plots for harmaline and fluoxetine. These would be best shown in the supplement.

Fig 3: Please add new panels to Fig 3 that plot phage induction vs ecA expression (from Fig S1). On quick observation, it appears that bleomycin may act via RecA at concentrations below 0.781 µg/mL (the lowest used for RecA-GFP expression). Please perform and add testing of bleomycin at the same low concentrations for recA expression as it was for phage induction (0.0003 µg/mL). Additionally a plot of phage induction vs RecA expression might better show this pattern.

Fig 4: What are the integration sites of each Mu-like phage (in E. coli and P. aeruginosa). This is trivial to determine with short read sequencing. Though it would not be an exhaustive survey, might shed light on induction mechanisms.

All MIC curves (for all compounds, and phage-host pairs) should be included as supplemental figures.

All MICs for the four compounds (or, as for berberine, the inferred MIC) and ciprofloxacin and mitomycin should be listed as a table for the ease of the reader.

Table S2: The table lacks any quantification of "effect on OD"

Line 11: "Prophages; dormant bacteriophage genomes integrated within the bacterial chromosome, play..." This semicolon should be a comma.

In some places the E. coli strain is named K12, and in others it is MG1655. Please be consistent.

Figure 2A: It is confusing and unclear to the average reader that one compound (Berberine) does not have a defined MIC (for a justifiable reason regarding spectrophotometry), but is a confirmed inducer. It should be made more explicit in the figure legend and text (Lines 192-194) how induction was quantified (plaque assay).

Fig 2B, 2C, and S2: The figure legend does not match the figures. There are no diagonal lines in the charts.

Fig 4 lacks a figure legend.

Nair et al present a very interesting analysis of bleomycin as an SOS-dependent prophage inducer. There are some low effort additions and/or new analyses that should be made, whether for clarity, completion, or just increasing the strength of the paper. It would especially benefit from trivial sequencing and analysis to determine Mu-like integration sites in the lysogens shown and their potential roles in induction or lack thereof. Together, this is an impressive manuscript that just requires a little more work.

A more thorough review of the literature is needed. There is no mention of previous work on bleomycin effects on phages, which is a known inducer though not commonly used. A quick Google search finds several that were not mentioned. Are your results consistent with previous findings? The conclusions that bleomycin should be used as a standard phage inducer in the lab would be better supported with greater literature analysis. Is your work and conclusion unique from these prior analyses?

Fig 2A: You say that “Regardless, lysogen was subjected to a range of Berberine concentrations and the greatest magnitude of induction (~ 2 -log increase) relative to baseline spontaneous induction levels was observed at 350 $\mu\text{g}/\text{mL}$ (Figure 2b).” You should plot the berberine concentration vs prophage induction (as you do for bleomycin in Fig 3). It would be relevant to add the same plots for harmaline and fluoxetine. These would be best shown in the supplement.

Fig 3: Please add new panels to Fig 3 that plot phage induction vs *ecA* expression (from Fig S1). On quick observation, it appears that bleomycin may act via RecA at concentrations below 0.781 $\mu\text{g}/\text{mL}$ (the lowest used for RecA-GFP expression). Please perform and add testing of bleomycin at the same low concentrations for *recA* expression as it was for phage induction (0.0003 $\mu\text{g}/\text{mL}$). Additionally a plot of phage induction vs RecA expression might better show this pattern.

Fig 4: What are the integration sites of each Mu-like phage (in *E. coli* and *P. aeruginosa*). This is trivial to determine with short read sequencing. Though it would not be an exhaustive survey, might shed light on induction mechanisms.

All MIC curves (for all compounds, and phage-host pairs) should be included as supplemental figures.

All MICs for the four compounds (or, as for berberine, the inferred MIC) and ciprofloxacin and mitomycin should be listed as a table for the ease of the reader.

Table S2: The table lacks any quantification of “effect on OD”

Line 11: “Prophages; dormant bacteriophage genomes integrated within the bacterial chromosome, play...” This semicolon should be a comma.

In some places the *E. coli* strain is named K12, and in others it is MG1655. Please be consistent.

Figure 2A: It is confusing and unclear to the average reader that one compound (Berberine) does not have a defined MIC (for a justifiable reason regarding spectrophotometry), but is a

confirmed inducer. It should be made more explicit in the figure legend and text (Lines 192-194) how induction was quantified (plaque assay).

Fig 2B, 2C, and S2: The figure legend does not match the figures. There are no diagonal lines in the charts.

Fig 4 lacks a figure legend.

Reviewer #1	Response
The manuscript "High-throughput screening reveals diverse lambdoid prophage inducers" shows the discovery and characterization of new potential prophage inducers (lambda-like). Overall, the manuscript is well organized with a clearly defined goal, which was the use of HTS to discover new prophage inducer compounds. Nevertheless, there are some questions/suggestions that were raised while reviewing the manuscript:	We thank the reviewers for the kind words.
Fig1-a) it is curious that a great portion of the compounds caused an increased relative growth on the HK97 lysogen. Do the authors have an explanation for this? It would be interesting to further explore these differences, which are apparently caused by the presence of this prophage.	We agree that this is very interesting – and went so far as to highlight an outlier that has a particularly pronounced effect (Benzoxiquine). We are currently exploring the effects of some of these compounds, but the results are preliminary – we can only say with certainty these are not inducers! As these are not central to the manuscript, we have not gone in further depth, although we have now added all the compound names to the raw data to simplify the process of readers pursuing compounds of interest.
In the ΔrecA mutant, it would be expected to have the same phage titer as the vehicle control. However, there is a different tendency with the compound treated culture yielding less phages than the no compound-treated. What would be the cause for this?	This is an excellent point. We have added a note; “Note that as the same concentrations are displayed as for Fig 2a, and the recA strains are often more susceptible to DNA-damaging compounds, we attribute the decreased titres for ciprofloxacin and mitomycin C to (phage-independent) impeded growth of the lysogen. “ When lower concentrations of the compound are used, effects are more similar to wildtype – but reporting a lower concentration that has no effect does not feel worthwhile (this can be true for any compound).
In lines 235 and 239 there is no reference to the physiological concentrations of Harmane and Bleomycin respectively. Despite the difficult estimation that the authors mention, it would be helpful to have some value to	We have added in some values to contextualize harmane and bleomycin levels. This now reads “ In a population-based study, the median

make it easier to compare with the results of this study. The authors also say that Bleomycin achieved "plasma levels". If these "levels" overlap with the concentration used, then they should definitely be mentioned.	blood concentration in controls was very low (2.7×10^{-10} g/mL) after habitual dietary intake (46). With both Berberine and Harmane, local concentrations in the gastrointestinal tract are likely to be substantially higher than systemic levels due the prevalence of Harmane in common diets and, for Berberine, the frequent supplement use, lack of standardized dosing and potential for repeated high dose exposure.”
Why did the authors used pitB for mRNA expression assays? And why use sulA as well?	We have added a note to the methods to explain this choice; “SulA was chosen to represent genes far deeper in the SOS pathway (56), while pitB was selected as it was known to be SOS-independent (57), and has been used as a control for similar reporter gene assays by other groups (58). “
In Fig4 there is no identification of the columns in the graphs (there are 2 per treatment)	We have corrected this, and a figure legend has been added.
In line 333 is referred P8 and P13 lysogens, but there is no reference to them in the graph	We have corrected this to refer to them by their phage names.
Small details:  • Table legends end without dot "." • In line 386 "OD" misses the wavelength used • Figure s2-b) "recA" in the graph should be italic 	We have corrected and clarified each of the points listed.
Reviewer #2	
This study reports the results of a high throughput screen of 3,921 compounds in an attempt to identify novel inducers of a typical E. coli lambdaoid prophage, HK97. This prophage has a canonical SOS-inducible cI-like repressor. The screen used differential growth inhibition of the lysogen by the tested compounds compared to an isogenic prophageless control strain, as a proxy readout of potential prophage inducers. 37 potential inducers were identified by the	We thank the reviewer for the kind words, and suggestions to improve the manuscript.

screen, and 13 were selected for further studies. Of these only 4 were verified as prophage inducers. Perhaps not surprisingly, all were dependent on the SOS response, showing no induction in a recA mutant background. Further characterization of the most potent prophage inducer, bleomycin, revealed a broader range and more potent induction than mitomycin C. Additionally, bleomycin was shown to be capable of prophage induction in certain uncharacterized Mu lysogens of E. coli and three different Pseudomonas aeruginosa lysogens. The results are clear and generally well presented, but attention to the following points would result in a much improved manuscript.	
1) The introduction needs a clear description of how induction occurs in lambdoid prophages via the cI repressor - this is completely lacking in the current manuscript. Also no reasoning is presented as to why compounds other than SOS inducers might be expected to induce the prophage.	This is correct in that we did not make this clear in the introduction. We have added the role of cI into the introduction (paragraph 2) as well as a reasoning as to why non-SOS inducing compounds may exist.
2) While figure 1b presents a pie chart of the type of compounds identified as primary hits - the majority of which proved not able to induce prophage - there are no details of the chemistry or variety of 3,921 compounds included in the initial screen.	We have now included a comprehensive list of the bioactive compound library within the raw data file.
3) Much previous research on identification and use of bleomycin as prophage inducer is downplayed, incorrectly suggesting only a single previous study (ref. 41, Elespuru and White, 1983). A Pubmed search for "bleomycin prophage" results in 14 hits, although all before 1986, studies from multiple different groups and covering not only E. coli and lambda, but prophages in Bacillus subtilis and Staphylococcus. A 1972 study abstract (article in Italian) indicated bleomycin to be more effective than	The reviewer is absolutely correct, and we completely agree. We have added in additional references (41-43) and further elaborated on previous reports establishing bleomycin as prophage inducer. We hope that we have conveyed that we are attempting to focus on the effectiveness of bleomycin as an inducer rather than its novelty.

mitomycin C at inducing lambda prophage.	
4) Consideration of serum concentrations of compounds as physiological effective concentrations is not very relevant to this in vitro laboratory assay development. Consider removing this.	We understand that serum concentrations are not required to convey the story, however due to the request of another reviewer to expand on some of this information we have added in some quantitative values but have also softened the language.
5) This study proclaims the novel identification of xenobiotics as prophage inducers, yet does not do any further characterization of these compounds or expand on their potential utility, focusing instead on more detailed characterization of a known lambdoid prophage inducer.	We have added in the following statement: “Moreover, the identification of these xenobiotics as prophage inducers highlights their potential utility as tools for probing phage-host dynamics and understanding how non-antibiotic compounds influence microbial communities in both clinical and environmental contexts”.
More minor points that should be addressed:  1. Description of methods need clarification Strains - it is not clear if the two E. coli strains are isogenic - i.e. if the HK97 lysogen is directly derived from HER1382; if not this needs stating. 2. Induction Assays. At what wavelength were OD readings taken? This is subsequently implied to be 600nm in the brief data analysis paragraph. Not clear why or where assay readings were taken every 30 mins or only at endpoint (endpoint only for SOS assays?) 3. HTS screening. Please provide more details on the selection of the bioactive compounds - is this an off the shelf set produced by CMCB for other purposes? Clarify assay description - it appears that compounds were first dispensed as 1 mM solution then 100x volume of culture was added - I had to reread this section multiple times to figure this out. 	 1. They are isogenic and we have tried to be more specific with the strain naming in Table 1. 2. We have added the specific OD wavelength under prophage induction assay. We agree with the reviewer that stating that reads were taken every 30 mins or only at endpoint could be confusing. We have adjusted it to be 18 hrs since only endpoint data is analysed and reported in this manuscript. 3. The bioactive compound library was curated by the CMCB facility located at McMaster University, Hamilton, ON. We have included a comprehensive list of the library as supplementary information. We agree that certain aspects of the assay description are confusing and have therefore simplified the HTS methods. The concentration of the compound library is 1mM thus adding 0.3 ul to a final volume of 30 ul results in a final compound concentration of 10 uM.
Results and discussion Figure 1. Not mentioned here are two also interesting groupings of compounds - 10 in	Figure 1: We agree with the reviewer that these compounds are interesting. We have pursued a few of them and are currently

the left-most region of upper left quadrant that appear to indicate compounds toxic to the lysogen. And a group of 30 or so in the upper right quadrant, that like benzoxyquine, appear to be beneficial to the lysogen. More could be made of the potential of this assay to identify interesting compounds in prophage biology beyond inducers. Figure 2. It is not clear or stated in the legend at what dose the compounds in b) and c) are used. Additionally, I am confused by the presence of multiple different no compound controls producing what appear to be significantly different phage titers. Comment on this variability and its potential effect. Figure 3. These are very busy graphs. Small dots are individual assays of the triplicate assays - not detailed? X axes are very crowded - consider removing unnecessary decimals for higher doses (we do not need 200.000 for 200 or 50.00 for 50 e.g.) Figure 4. Again, discuss the variation in no compound controls across experiments as an uncontrolled variable. Compound concentration needs explicitly stating. Temper discussion of "novelty" of bleomycin induction in non lambda and non-E. coli organisms with reference to above cited PubMed articles for other phage and bacteria.	investigating their effects. However since they are not essential to the story, we have not included any preliminary findings. Figure 2: Concentrations used for the Figure 2b) and 2c) are listed within the text below. Regarding the variability of the no compound controls, we noticed that prophage induction levels in the absence of a stressor varied from assay to assay. To account for this variability, we kept no compound controls separate for each compound. Additionally, despite the biological variability we are still able to see significant difference between no compound and compound added conditions making our conclusions, if anything, more robust. Figure 3: Induction levels are averaged across 3 biological replicates with each black dot representing a replicate. We have added this clarification into the Figure 3 caption as well as in other figure captions. We have also simplified Figure 3 x-axes by removing unnecessary decimals. Figure 4: The explanation for the variation here is the same as above. Compound concentrations have been added to the figure caption but are also listed in the text below.
Figure S1. Explain the use of a SulA reporter strain. Edit legend to show control is not MG1655 but a pitB-GFP fusion? Figure S3 - explain anomaly of MIC levels shown versus figure 3 - here Cipro is 2.5 µg/ml (0.063 in fig 3) Bleomycin is 200 µg/ml (3.125 in fig 3). Table S2. Also identify other compounds (another color?) selected that did not induce prophage on subsequent testing	Figure S1: We included sulA to capture downstream SOS activation, as sulA expression occurs predominantly in the later stages of the SOS response and/or in the presence of substantial DNA damage. This has been clarified in the methods. We have edited Figure S3 legend to state values were normalized to pitB-GFP. Figure S3: We think the reviewer is referring to Figure S1 (which has been changed to S3 in the manuscript) here. The MIC levels are different to those reported in Figure 3 since a different E. coli strain is being used here.

	Fluorescent assays used E. coli MG1655 (we have clarified this in the methods section under SOS gene expression assay) whereas prophage induction experiments use E. coli K12. Table S2 (now Table S3): We have highlighted compounds which we were unable to validate as prophage inducers in grey.
Reviewer #3	
Nair et al present a very interesting analysis of bleomycin as an SOS-dependent prophage inducer. There are some low effort additions and/or new analyses that should be made, whether for clarity, completion, or just increasing the strength of the paper. It would especially benefit from trivial sequencing and analysis to determine Mu-like integration sites in the lysogens shown and their potential roles in induction or lack thereof. Together, this is an impressive manuscript that just requires a little more work.	We thank the reviewer, and have added the additional analysis requested!
A more thorough review of the literature is needed. There is no mention of previous work on bleomycin effects on phages, which is a known inducer though not commonly used. A quick Google search finds several that were not mentioned. Are your results consistent with previous findings? The conclusions that bleomycin should be used as a standard phage inducer in the lab would be better supported with greater literature analysis. Is your work and conclusion unique from these prior analyses?	We completely agree with the reviewer in that we have overlooked previous reports validating bleomycin as a prophage inducer. We have added in references 41-43 and further elaborated on previous work on bleomycin's effects on phages as well as how our results compare. We hope that we have conveyed that we are attempting to focus on the effectiveness of bleomycin as an inducer rather than its novelty.
Fig 2A: You say that "Regardless, lysogen was subjected to a range of Berberine concentrations and the greatest magnitude of induction (~2-log increase) relative to baseline spontaneous induction levels was observed at 350 µg/mL (Figure 2b)." You should plot the berberine concentration vs prophage induction (as you do for bleomycin	We thank the reviewer for this suggestion and have added in Figure S2.

in Fig 3). It would be relevant to add the same plots for harmaline and fluoxetine. These would be best shown in the supplement.	
Fig 3: Please add new panels to Fig 3 that plot phage induction vs recA expression (from Fig S1). On quick observation, it appears that bleomycin may act via RecA at concentrations below 0.781 ug/mL (the lowest used for RecA-GFP expression). Please perform and add testing of bleomycin at the same low concentrations for recA expression as it was for phage induction (0.0003 ug/mL). Additionally a plot of phage induction vs RecA expression might better show this pattern.	We attempted this analysis and correlation was weak but to further explore the reviewer's comments we expanded the recA reporter assays to include lower concentrations.
Fig 4: What are the integration sites of each Mu-like phage (in E. coli and P. aeruginosa). This is trivial to determine with short read sequencing. Though it would not be an exhaustive survey, might shed light on induction mechanisms.	We have elaborated on Mu integration sites within the bacterial strains table under the methods.
All MIC curves (for all compounds, and phage-host pairs) should be included as supplemental figures. All MICs for the four compounds (or, as for berberine, the inferred MIC) and ciprofloxacin and mitomycin should be listed as a table for the ease of the reader.	We thank the reviewer for this suggestion and have added in Figure S1 as well as Table S1.
Table S2: The table lacks any quantification of "effect on OD"	We have just ordered the list according to OD effects but did not feel the values to be relevant to the story. OD effect is a proxy for phage titre, which we measure directly later anyway.
Line 11: "Prophages; dormant bacteriophage genomes integrated within the bacterial chromosome, play..." This semicolon should be a comma.	This has been corrected.
In some places the E. coli strain is named K12, and in others it is MG1655. Please be	While we understand the reviewer's confusion, these strains are not the same. To

consistent.	provide additional clarity we have specified the strain name of the K12 strain used according to the source library. HER1382 and MG1655 are K12 derivatives obtained from different sources.
Figure 2A: It is confusing and unclear to the average reader that one compound (Berberine) does not have a defined MIC (for a justifiable reason regarding spectrophotometry), but is a confirmed inducer. It should be made more explicit in the figure legend and text (Lines 192-194) how induction was quantified (plaque assay).	We have clarified this within the text.
Fig 2B, 2C, and S2: The figure legend does not match the figures. There are no diagonal lines in the charts.	The no compound bars in Figure 2B, 2C and PCR H₂O are textured with diagonal lines which is what is reflected in the legend.
Fig 4 lacks a figure legend.	This has been corrected and a figure legend has been added.

Re: Spectrum01707-25R1 (High-throughput screening reveals diverse lambdoid prophage inducers)

Dear Dr. Alexander P Hynes:

Thank you for submitting the revised version of the manuscript. Please address the minor comments raised by the reviewer.

Revision Guidelines

Sincerely,
Diana Pires
Editor
Microbiology Spectrum

Reviewer #4 (Comments for the Author):

Nair et al. performed a screen for prophage inducers in E. coli lambdoid phages and identified four compounds, with bleomycin emerging as a particularly potent and broad-spectrum inducer. The current version of the manuscript is clear and introduces new prophage inducers.

Although other prophages were tested, these were limited to another prophage from the same species and one prophage from Pseudomonas. This remains a relatively narrow scope, so the claims should be reframed to avoid speculation and to avoid overstating the broader effect.

Minor:

These two sentences in introduction are important but repetitive: " Most studies rely on synthetic agents like Mitomycin C, and Ciprofloxacin, as well as Ultraviolet (UV) light, to stimulate prophage induction in model systems like lambda, a temperate phage of Escherichia coli (6-8). Exposure to these strongly and reproducibly induces the bacterial SOS response, which in turn triggers prophage induction (8-10). While effective, this approach has become the default across studies, potentially biasing our view of phage-host interactions and limiting the discovery of alternative inducers that may be more relevant to natural environments or clinical contexts.." And in "This is especially important given the field's historical reliance on canonical inducers such as Mitomycin C and Ciprofloxacin, which may obscure other ecologically or clinically relevant triggers.". Both sentences emphasize the field's reliance on canonical inducers such as Mitomycin C and Ciprofloxacin. Since the first sentence already develops this point in detail, the second risks sounding redundant. I suggest reframing the second to highlight a different nuance (e.g., the ecological/clinical implications of this reliance, or the specific knowledge gap the present study aims to address).

Table 1 - italicize gene names

Reviewer #4	Response
Nair et al. performed a screen for prophage inducers in E. coli lambdaoid phages and identified four compounds, with bleomycin emerging as a particularly potent and broad-spectrum inducer. The current version of the manuscript is clear and introduces new prophage inducers.	We thank the reviewers for the kind words.
Although other prophages were tested, these were limited to another prophage from the same species and one prophage from Pseudomonas. This remains a relatively narrow scope, so the claims should be reframed to avoid speculation and to avoid overstating the broader effect.	The language has been softened somewhat, although we will note that we did test three different Pseudomonas prophages. The abstract now reads: “However, bleomycin, an antineoplastic antibiotic, demonstrated broad-spectrum and potent prophage induction exceeding standard inducers, with activity validated across multiple phage-host pairings.” The transition to non HK97-phages now reads “To determine whether this represents a generalizable phenomenon, we tested Bleomycin’s ability to induce other phages, including in other bacteria.” And the final section of the Pseudomonas work now reads “while Bleomycin induced all three lysogens and resulted in higher magnitudes of induction. This further emphasizes the compound’s potency and suggests generalizability across different bacterial host and their prophages, as has been demonstrated for Ciprofloxacin and Mitomycin C.” to highlight that the claim of broader efficacy is not particularly extraordinary.
These two sentences in introduction are importante but repetitive: " Most studies rely on synthetic agents like Mitomycin C, and Ciprofloxacin, as well as Ultraviolet (UV) light, to stimulate prophage induction in model systems like lambda, a temperate phage of Escherichia coli (6-8). Exposure to	We agree, and rather than pad the manuscript, we have simply deleted the second – redundant – claim.

these strongly and reproducibly induces the bacterial SOS response, which in turn triggers prophage induction (8-10). While effective, this approach has become the default across studies, potentially biasing our view of phage-host interactions and limiting the discovery of alternative inducers that may be more relevant to natural environments or clinical contexts.." And in "This is especially important given the field's historical reliance on canonical inducers such as Mitomycin C and Ciprofloxacin, which may obscure other ecologically or clinically relevant triggers.". Both sentences emphasize the field's reliance on canonical inducers such as Mitomycin C and Ciprofloxacin. Since the first sentence already develops this point in detail, the second risks sounding redundant. I suggest reframing the second to highlight a different nuance (e.g., the ecological/clinical implications of this reliance, or the specific knowledge gap the present study aims to address).	
Table 1 - italicize gene names	The gene names have now been correctly italicized.

Re: Spectrum01707-25R2 (High-throughput screening reveals diverse lambdoid prophage inducers)

Dear Dr. Alexander P Hynes:

I am glad to inform that your manuscript has been accepted, and I am forwarding it to the ASM production staff for publication. Your paper will first be checked to make sure all elements meet the technical requirements. ASM staff will contact you if anything needs to be revised before copyediting and production can begin. Otherwise, you will be notified when your proofs are ready to be viewed.

Sincerely,
Diana Pires
Editor
Microbiology Spectrum